# An Analysis of the Possible Migration Routes of *Oedaleus decorus asiaticus* Bey-Bienko (Orthoptera: Acrididae) from Mongolia to China

**DOI:** 10.3390/insects13010072

**Published:** 2022-01-10

**Authors:** Yunping Wang, Shuang Li, Guilin Du, Gao Hu, Yunhui Zhang, Xiongbing Tu, Zehua Zhang

**Affiliations:** 1State Key Laboratory of Biology of Plant Diseases and Insect Pests, Institute of Plant Protection, Chinese Academy of Agricultural Sciences, Beijing 100125, China; 2017102066@njau.edu.cn (Y.W.); sclishuang61@163.com (S.L.); zhangzehua@caas.cn (Z.Z.); 2Institute of Applied Agricultural Micro-Organisms, Jiangxi Academy of Agricultural Science, Nanchang 330008, China; 3College of Plant Protection, Nanjing Agricultural University, Nanjing 210095, China; hugao@njau.edu.cn; 4Scientific Observation and Experimental Station of Pests in Xilin Gol Rangeland, Institute of Plant Protection, Chinese Academy of Agricultural Sciences, Xilinhot 026000, China; 5National Animal Husbandry Service, Ministry of Agriculture and Rural Affairs, Beijing 100125, China; caasdgl@163.com

**Keywords:** grasshoppers, migration regularity, meteorology, Beijing, Mongolia, long distance flight, air current, weather model

## Abstract

**Simple Summary:**

Airflow is very important for the long-distance migration of *O. decorus asiaticus*, and wind shear, in particular, is the main factor related to forced landing. Analyzing the weather records, we found that the northwest wind prevailed when the population invaded. Specifically, from July to August, a large number of emerging adults appeared in the source areas of Mongolia, and the large-scale northwest wind was the decisive condition for the successful long-distance migration of *O. decorus asiaticus*. The species has a strong migratory ability, flying along the airflow for several nights. If the northwest air current meets the southwest warm current going north, a large number of *O. decorus asiaticus* will drop due to wind shear, and then a major outbreak will occur. Analysis of the source of the insects shows that the *O. decorus asiaticus* break outs in China may have originated from Mongolia. They were brought into China by the southerly airflow at night, and they likely made a forced landing in Beijing due to wind shear, sinking airflow, rainfall and other reasons. In summary, through analysis of the insect’s prevalence and the meteorological conditions in Mongolia, we can provide a basis for predicting the occurrence of *O. decorus asiaticus* in China.

**Abstract:**

*Oedaleus decorus asiaticus* (Bey-Bienko) is a destructive pest in grasslands and adjacent farmland in northern China, Mongolia, and other countries in Asia. It has been supposed that this insect pest can migrate a long distance and then induce huge damages, however, the migration mechanism is still unrevealed. The current study uses insect light trap data from Yanqing (Beijing), together with regional meteorological data to determine how air flow contributes to the long-distance migration of *O. decorus asiaticus*. Our results indicate that sinking airflow is the main factor leading to the insects’ forced landing, and the prevailing northwest wind was associated with *O. decorus asiaticus* taking off in the northwest and moving southward with the airflow from July to September. Meanwhile, the insects have a strong migratory ability, flying along the airflow for several nights. Thus, when the airflow from the northwest met the northward-moving warm current from the southwest, a large number of insects were dropped due to sinking airflow, resulting in a large outbreak. Our simulations suggest that the source of the grasshoppers involved in these outbreaks during early 2000s in northern China probably is in Mongolia, and all evidence indicates that there are two important immigrant routes for *O. decorus asiaticus* migration from Mongolia to Beijing. These findings improves our understanding of the factors guiding *O. decorus asiaticus* migration, providing valuable information to reduce outbreaks in China that have origins from outside the country.

## 1. Introduction

Outbreaks of grasshoppers have had increasingly serious impacts on human activity in China over the past millennia. *Oedaleus decorus asiaticus* (*O. decorus asiaticus*) is an important pest affecting grasslands and adjacent farmland, and has caused huge damage to the livestock industry and ecological environment [1,2]. In China, *O. decorus asiaticus* mainly appears in Inner Mongolia, in typical grassland and desert steppe regions from the central and west of the Xilinguole League to the east of Ordos City [3]. During early July of 2002–2004, there were several sudden outbreaks of *O. decorus asiaticus* in northern China, including Chifeng City, Duolun County, Zhangjiakou City, Chengde City and Beijing [4]. We know that only adult insects have strong flight ability [5], with a maximum flight distance of 350 km in one night [6,7]. However, we found that the local population consists only of the fifth stage hoppers [4], and these cannot migrate over a long distance. Thus, we speculated that the source of *O. decorus asiaticus* involved in these outbreaks is likely to be an invasive population. However, the migration mechanism is still unrevealed.

Similar to factors affecting other migratory insects’ flight ability and migration regularity [8], airflow has a strong influence on the success of *O. decorus asiaticus*’ long-distance migration due to its inherently limited flight ability. Therefore, variations in seasonal airflow can have a critical effect on the annual migration of insects [9,10]. Weather factors, such as strong convergent airflows, can cause the forced landing of migratory individuals which terminates their flight, and can even lead to a large number of individuals landing in a concentrated manner, causing a large regional outbreak [11,12,13,14]. Thus, the causes of *O. decorus asiaticus* outbreaks and their migration patterns are closely related to climate factors. Characteristics of the weather can induce migration and have important impacts on flight. In particular, the insects can only take off on a large scale under suitable temperature and humidity conditions [15,16,17,18,19]. Swarming locusts usually move downwind so that the wind will take them to their destination [20]; suitable airflow takes them to a green vegetation habitat, providing them with food and new spawning opportunities.

Beijing is not a suitable habitat for the *O. decorus asiaticus*, and studies show that they cannot overwinter in Beijing [4]. However, there are reports of the occurrence of *O. decorus asiaticus* in this area every year, indicating that there is some immigration from other locations. In this study, we use Yanqing light trap data in combination with reanalyzed meteorological data available from the National Centers for Environmental Prediction (NCEP; United States of America) to develop a three-dimensional insect trajectory analysis. We used the Weather Research and Forecast model (WRF 3.9 model) [21] together with GrADS meteorological graphics software to simulate and visualize the migration paths of the *O. decorus asiaticus* migration into the Beijing area, and to identify the meteorological factors affecting these paths. Our study helps to clarify the migration regularity of the *O. decorus asiaticus*, and to determine the source and migration route of the small population of *O. decorus asiaticus* in Beijing.

## 2. Materials and Methods

### 2.1. Data Sources

#### 2.1.1. Insect Information

Data on flying insects attracted by a light lamp in the Yanqing observation station, Institute of Plant Protection, Chinese Academy of Agricultural Sciences, provided the data regarding light trapping of flying insects in the Yanqing area. In the Yanqing District of Beijing (40.46° N, 115.98° E), adult insects were captured with searchlights every night from May to September in 2012–2017. The running time of the lamp was 20:00–05:00, except for instances of power failure or heavy rain. A vertically illuminated searchlight (dk.z.j1000b/t, made in Shanghai, China) was placed on a platform approximately 1.5 m a.s.l. and was used to trap high-altitude migrants flying overhead up to approximately 500 m above ground level. The captured insects were collected in a nylon net bag, which is placed under the light trap and replaced every hour during the night. Before identification, the insects were frozen at −20 °C for 4 h, and then the species and number of insects in the bag are recorded.

#### 2.1.2. Meteorological Data

Meteorological data were obtained from the global reanalysis data (Final Analysis, FNL) of the NCEP and the National Center for Atmospheric Research (NCAR; United States of America). Data are prepared on grids with a horizontal resolution of 1.0° × 1.0° and each analysis includes global meteorological data over a time interval of 6 h.

### 2.2. Analysis of Three-Dimensional Trajectory and Atmospheric Circulation Fields Based on the WRF Model

#### 2.2.1. WRF Model Initial Conditions

To develop our quantitative meteorological simulation, we used a terrain resolution of 2′, covering the world’s Moderate Resolution Imaging Spectroradiometer (MODIS) and Terrain Gravity Wave Drag by Orography (GWDO) data. The NCEP FNL Global Tropospheric Analysis data was used to inform the initial field data and boundary conditions for the WRF model. After meteorological data are inputted, the model outputs an hourly forecast with 30 km × 30 km grid spacing which provides the conditions to allow us to calculate potential migration trajectories. The wind field data used in the trajectory calculation was derived from the WRF model, which is a new generation of mesoscale numerical weather prediction system. The model provides a high-resolution atmospheric background for the trajectory analysis of this study. The simulation area and parameter settings of the WRF model are shown in Table 1.

#### 2.2.2. Analysis of the Atmospheric Circulation Background Fields of *O. decorus asiaticus*

Using the results of the WRF simulation and the pest situation records from agricultural stations, we selected the atmospheric field of the average nighttime high altitude flow field from the peak day of pest reduction, and selected the flow field map at 800 m altitude in the Beijing area. The weather background field of the *O. decorus asiaticus* flying at night was visualized using GrADS 2.1, and the influence of upper air flow on the migration and landing of *O. decorus asiaticus* population was analyzed.

To identify the potential sources of *O. decorus asiaticus* arriving in the Yanqing area, we simulated potential migratory routes for five nights backwards in time from each date that *O. decorus asiaticus* were captured in Yanqing. The trajectories were estimated with the following assumptions: (a) the migration speed at which the *O. decorus asiaticus* travels by air is the vector sum of the wind speed and its own flight speed, and its own flight speed is about 3 m/s [22,23,24]; (b) the migration time of the *O. decorus asiaticus* in the first night is from 20:00 p.m. on the night of take-off to their captured time, and the remaining four nights are from 20:00 p.m. to 3:00 a.m. of the next day, lasting for 7 h [4]; (c) *O. decorus asiaticus* migrates at an altitude of 500–1500 m above the ground [25]. Because the average altitude of the Mongolian Plateau is more than 1000 m, eight flight altitudes were used in this study: 750, 1000, 1250, 1500, 1750, 2000, 2250 and 2500 m above sea level [26]; (d) *O. decorus asiaticus* can migrate continuously for multiple nights. As mentioned above, the trajectory was estimated for the five consecutive nights prior to the night when the *O. decorus asiaticus* was captured, the landing points of the first night will serve as the departure points of the second night, and each departure point will have eight flying altitudes. In the next night, the end point of the trajectory of one night’s travel was used as the starting point for the next day’s take-off [24,27].

The trajectory simulation was carried out according to the high-altitude horizontal flow field and the above-mentioned biological parameters. In addition, we screened potential and effective migration trajectories based on characteristics, such as terrain, distribution area, and other biological characteristics [28]. The criteria for effective trajectories are as follows: (1) the time at the end of the trajectory should be in accordance with the take-off rhythm of the *O. decorus asiaticus*; (2) there must be plants suitable for the feeding of *O. decorus asiaticus* at the end of the trajectory; (3) *O. decorus asiaticus* occurs in the trajectory termination area and can provide an effective source of emigration. Only when the flying altitude of *O. decorus asiaticus* is lower than the eight sea-level altitudes used (750, 1000, 1250, 1500, 1750, 2000, 2250 and 2500 m), can its flight trajectory can be maintained for five consecutive nights. Some tracks ended in less than five nights due to terrain factors (Figure 1).

#### 2.2.3. The Choice of Moving in Peak Day

By screening the trapping data from 2012 to 2017 and analyzing the start date of the immigration process of *O. decorus asiaticus*, we chose all the nights when the grasshoppers were captured as peak days, except for the ones that were captured between 8 p.m. to 9 p.m., because the grasshoppers taking off at 8 p.m. and landing at that time did not immigrate [19]. In total, we obtained 69 peak days (the date with trapped *O. decorus asiaticus*).

## 3. Results

### 3.1. Analysis of Population Dynamics under Light

The data of light traps from 2012 to 2017 showed that the period of light traps in Yanqing was from June and September and the peak period of light traps in Yanqing was from July and August, and longest trap time is 2017, lasting 31 days. In addition, while in recent years there were not many *O. decorus asiaticus* trapped by light traps in Yanqing from 2012 to 2017. The number was significantly higher in 2017 than in other years, with 566 trapped, with the largest number of *O. decorus asiaticus* captured from 8:00 p.m. on 15 August to 5:00 a.m. on 16 August 2017 (142). There was almost no light-captured grasshoppers in other months. *O. decorus asiaticus* hatch in May, and gradually emerge in July every year. Therefore, there is only successful light trapping of *O. decorus asiaticus* in Yanqing from July to August (Table 2).

### 3.2. Analysis of the Possible Origin of Migrating O. decorus asiaticus Adults

From 2012 to July–September 2017, various numbers of *O. decorus asiaticus* were trapped. With the date of the night that the trap recorded as the starting date, the backtracking trajectory analysis was carried out for five nights (Figure 1) in order to identify the potential source area of the *O. decorus asiaticus*. The results of this trajectory analysis demonstrated that a large number of *O. decorus asiaticus* originated from Hebei and Shanxi in China, and most originated from Inner Mongolia, suggesting that *O. decorus asiaticus* invade Hebei, Beijing and other places from Inner Mongolia. The trajectories estimated from the 2017 and 2012 data indicated that if the conditions are suitable, *O. decorus asiaticus* can also migrate long distances from Mongolia, entering Chinese territory, and arriving in Yanqing after several nights.

The results of the trajectory analyses showed that while some trajectories terminated early (i.e., were less than five nights), most of the other trajectories come from the northwest, except for a small number of tracks from Southwest China. As can be seen in Figure 1, in 2012–2017, some trajectories entered Hebei from Inner Mongolia, and then continued into Yanqing, while others came directly from Mongolia; in 2014, the trajectories entered Hebei from Shanxi, Shanxi and Gansu, and then proceeded to Yanqing. There are some trajectories came from Heilongjiang in August 2015, and some from Shandong in 2017. Trajectories from 2012, 2015 and 2016 all indicated paths from Inner Mongolia which entered Hebei after several consecutive nights and then arrived in Yanqing. In 2017, in addition to tracks from Inner Mongolia, Shanxi and Hebei, a number of trajectories also originated in Mongolia and arrived in Yanqing via Hebei, and arrived in China one night later, entering Hebei through Mongolia, and finally arriving in Yanqing (Figure 1).

Of the calculated 14,297 trajectories, 12,819 (89.67%) were valid. There were 1478 trajectories that were terminated due to the high altitude of Inner Mongolia, accounting for 10.33% of the total trajectories (Table 3). Excluding the forcibly terminated trajectory landing points and invalid *O. decorus asiaticus* source landing points, there were a total of 12,833 effective trajectory landing points, which are mainly distributed in Inner Mongolia, Shanxi, and Hebei (Figure 1).

These effective trajectories are mainly from Shanxi, Hebei, and Inner Mongolia, with a total of 10,382 (72.62%). Among them, the most effective trajectories were from Hebei with 5468, followed by Shanxi, Inner Mongolia, and Beijing, with 3112, 1802 and 1314, respectively. A small number of them came from Henan (414/2.90%), Liaoning (271/1.90%), Shaanxi (257/1.80%), Mongolia (49/0.34%), Tianjin (39/0.27%), Heilongjiang (39/0.27%), Gansu (20/0.14%), Jilin (10/0.07%), Shandong (9/0.06%), Chongqing (4/0.03%), Hubei (10/0.07%) and the sea surface (6/0.04%) (Table 3 and Table 4).

In addition, 1478 trajectories were invalid trajectories and were forced to stop because of moving beyond the simulated domain. Of these, 951 terminated in Hebei (65.0%), 263 in Beijing (18.0%), 195 in Shanxi (13.3%), and the remaining 49 in Inner Mongolia (3.3%) (Table 3 and Table 4).

The analysis of the backtracking trajectories of the *O. decorus asiaticus* on the peak days of migration from 2012 to 2017 showed that the *O. decorus asiaticus* sources in Yanqing from July to September are mainly distributed in Hebei, Shanxi, Inner Mongolia, and Beijing, with a small number of *O. decorus asiaticus* sources distributed in Henan, Liaoning, Shaanxi, Mongolia, Tianjin, Heilongjiang, Gansu, Jilin, and Shandong in China.

### 3.3. Landing Mechanisms

#### 3.3.1. Landing Mechanisms Affecting *O. decorus asiaticus* Migration in Yanqing, China

The landing mechanisms affecting *O. decorus asiaticus* on the 69 migration peak days from 2012 to 2017 are summarized in Appendix A. Among them, landing on 24 days was associated with wind shear, 31 days were associated with sinking airflow, and 25 days were related to precipitation. It can be seen from the table that precipitation and sinking airflow are the key important factors for the landing of the *O. decorus asiaticus* in the Yanqing area.

#### 3.3.2. Case Analysis of Moving into the Peak on 8 July 2017

On 6 July 2017, a Mongolian cyclone formed at a center of low pressure in central Mongolia. By 8 July, the center of this cyclone had moved northeast, resulting in northwest wind control over Mongolia and Inner Mongolia. On 8 July 2017, at the height of 925 hPa at night in Yanqing District, affected by the Mongolian cyclone, a northwest air flow greater than 3 m/s passed through the middle of Inner Mongolia to the east of Mongolia. A northwest airflow of greater than 6 m/s passed through the central part of Inner Mongolia to the eastern part of Mongolia. Under these conditions, adult insects from Mongolia would be able to travel southward with the wind and successfully arrive in Inner Mongolia and Hebei, Yanqing and other places in China.

At this time, in the south, the cold air flow met a warm air flow greater than 6 m/s over Hebei and Beijing, resulting in wind shear (Figure 2a) and a strong vertical airflow disturbance occurred on 7 July to 13 July. The speed of the sinking airflow was over 0.4 Pa/s (Figure 2b), which is sufficient to cause *O. decorus asiaticus* to land in the Yanqing area, with a resultant light trap peak in Yanqing at night.

This sequence of events is supported by evidence from our backward trajectory simulation for 8 July 2017, which captured the effect of the Mongolian cyclone: the simulated trajectories came from the northwest, turned to the northeast, and the grasshopper could then enter Yanqing after passing through the central part of Inner Mongolia from outside Mongolia (Figure 1).

### 3.4. Dynamic Analysis of O. decorus asiaticus Migration in 2002

Figure 3 illustrates how the large worms present in Chengde, Zhangjiakou, and Beijing were caused by the Mongolian cyclone center, which was at 925 hPa on 10 July in the Mongolian Plateau. Moving east and reaching eastern Mongolia, a large area of cold southward air appeared in northern China, causing a prevailing northwest wind in Inner Mongolia, northern Hebei and other areas. At this time, the flow field conditions were conducive to *O. decorus asiaticus* migrating southward and entering China. At the same time, the cold air from the south and the warm and humid air travelling southwest from the north crossed over northern Hebei, Beijing, and eastern Inner Mongolia, forming a long and narrow convergence zone and leading to strong convective weather, which could lead to large-scale *O. decorus asiaticus* landing. These flow field conditions lasted for three days, and the strong warm air flow prevented the *O. decorus asiaticus* from continuing to move south; instead, a large number landed in Chengde, Zhangjiakou, Beijing and nearby areas, causing a large number of *O. decorus asiaticus* to emerge in these areas on 10–12 July.

### 3.5. Dynamic Analysis of O. decorus asiaticus Migration in 2003

According to the records of Beijing Academy of Agricultural Sciences, there was a large-scale migration of *O. decorus asiaticus* in Xisu Banner in Inner Mongolia, China on 18 July 2003. The swarm entered Hohhot in China from 30 to 31 July, and a large number then entered Yanqing on 3 August. As the average altitude of Inner Mongolia is above 1000 m, the night wind field was visualized at an altitude of 1500 m (850 hPa). Combined with the analysis of the wind field at night, the airflow conditions at this time were suitable for the *O. decorus asiaticus* to move into China. On 18 July 2003, when the northern cold air flow was moving southward (Figure 4a), the migratory *O. decorus asiaticus* entering Xisu Banner were likely to be the Mongolian *O. decorus asiaticus* migrating downwind and entering Chinese territory, leading to a large-scale migration of adult insects appearing in Xisu Banner on 18 July 2003. On 30 July, the grasshoppers moved to the north of Hohhot, and some of them moved to the south of Hohhot (Figure 4b). On 3 August, following the northwest airflow into Yanqing City, a large number of *O. decorus asiaticus* were found in Yanqing City.

## 4. Discussion

*Oedaleus decorus asiaticus* is becoming an increasingly serious pest in the agricultural and pastoral ecotone in northern China. Due to its ability to cause a wide range of pastureland damage and engage in long distance migration, it is a difficult problem to monitor and control. Therefore, an understanding of their migration regularity is crucially important for the forecasting and control of many acridoid pests [29]. Trajectory regression analysis is one of the most common and effective methods to determine the possible source of migratory insects. It has been widely used in trajectory simulation analysis of migratory pests, such as *Mythimna separata* (Walker) (Noctuidae), *Nilaparvata lugens* (Stål) (Delphacidae), *Cnaphalocrocis medialis* (Guenee) (Crambidae), and *Spodoptera frugiperda* (Smith) (Noctuidae) [30,31,32,33]. Because insects are small and have limited flight capabilities, their migration ability cannot be compared with that of mammals, birds and other large animals. However, studies on the migration strategies of Noctuidae adults, such as *Helicoverpa armigera* (Hubner) and *Argyrogramma agnata* (Staudinger), have shown that insects with the ability to migrate autonomously can drift farther with the wind than inert particles in the air. This reflects insects’ use of an adaptive migration strategy related to the wind temperature field of the atmospheric boundary layer [9,21]. That is, to achieve long-distance migration, insects often use suitable wind fields to help them migrate [34].

There was no initial record of *O. decorus asiaticus* distribution in Beijing. In 2002, Jiang Xiang et al. kept the collected *O. decorus asiaticus* in the laboratory and found that while some of them could survive for a short time, the long-term survival rate of adults was extremely low, and the number of eggs laid was very small [4]. This finding indicated that *O. decorus asiaticus* were not coming from a local source, suggesting the likelihood of long-distance migration. That is, it seemed likely that most of the *O. decorus asiaticus* caught in Beijing migrated from outside populations.

In order to explore its migration process, we assumed a flight capability of 3.0 m/s [25] when simulating the migration trajectory of the *O. decorus asiaticus*, and used the eight flight altitude parameters of 750, 1000, 1250, 1500, 1750, 2000, 2250, and 2500 m. Through trajectory analysis, we concluded that the flight altitude is more than 1250 m. This is because the average altitude of Inner Mongolia is above 1000 m, so lower altitude trajectories will terminate in fewer than five nights, before the insects could have reached Beijing. The parameters of take-off and landing time, continuous flight time, migration times and flight altitude were set to accurately simulate the origin of *O. decorus asiaticus* in the Beijing area. According to the trajectory of light-trapped adults from 2012–2017, it is clear that most of the *O. decorus asiaticus* sources are in Hebei, Shanxi, Inner Mongolia, and Beijing, with a smaller proportion originating in Henan, Liaoning, and Shaanxi. The eastern part of Mongolia can also be the source of *O. decorus asiaticus* in China. Since there is no *O. asiaticus* distribution record in Chongqing, Hubei and the sea surface, the landing point is invalid [3]. The migration of *O. decorus asiaticus* does not involve completely passively drifting with the wind, but to a certain extent the insects autonomously choose to travel with the wind, migrating when conditions are suitable [19]. Insects migrating with the wind will intensively land under meteorological conditions, such as heavy rainfall, sinking air currents, and combined wind directions. They will also actively land when their energy materials are exhausted or the temperature drops below their flight threshold [34,35,36]. Among these factors, precipitation and sinking airflow are two important meteorological factors that affect the landing of aerial insect clusters [13]. For example, Guanheng Jiang et al. analyzed the meteorological factors during the 67 northward migrations and 15 southward migrations of the brown planthopper from 1977 to 1978, and found that precipitation and sinking airflow were the main meteorological factors affecting the large-scale landing of the brown planthopper [37]. Bao Yunxuan et al. analyzed the migration process of *Sogatella furcifera* (Horváth) (Delphacidae) that appeared on 10–11 July 2003 and found that rainfall was the direct cause of the concentrated landing of this planthopper species [38]. In the current study, we analyzed likely landing mechanisms on the 69 peak days of *O. decorus asiaticus* migration into Yanqing area in July to September from 2012–2017 (Appendix A); 24 times were related to wind shear and 31 times were related to the down draft, and 25 times were related to precipitation. Previous studies have shown that wind shear can cause large-scale landings of migratory insects [36,37,38]. In this study, 24 landings of *O. decorus asiaticus* were related to wind shear. Wind shear causes changes in the wind speed and direction of the airflow, and the sudden change in temperature can interrupt the southward migration of the *O. decorus asiaticus*, instigating a concentrated landing. Cold air from Siberia flows southward, and meets the warm air coming from the south, forming a long and narrow convergence zone. The wind shear thus formed becomes the main factor that causes *O. decorus asiaticus* to land. The Yanqing area belongs to a temperate monsoon climate, which is characterized by hot and rainy summers and cold and dry winters. Our results showed that the 25 selected immigration peaks all appeared in July or August, synchronized with the rainy season. Rainfall is an important factor influencing the concentration and decline of populations of *O. decorus asiaticus* in Yanqing. Because the terrain of Beijing and Hebei is high in the northwest and low in the southeast, gradually descending from northwest to southeast, the vertical disturbance caused by the decline of the terrain during the movement of the airflow also influences *O. decorus asiaticus* landing patterns. In addition, there are 17 landing times of *O. decorus asiaticus* which were not found to be related to rainfall, wind shear and downdraft, which may have occurred because their energy materials were exhausted, but the specific reasons need further investigation.

The genus *Oedaleus* originated in Ethiopia, Africa. Most of its species are closely related to locusts [39,40,41]. Based on mitochondrial DNA analysis, the genus *Oedaleus* may be the ancestor of locusts [41,42]. Although *O. decorus asiaticus* is a grasshopper, it also has the ability of locust’s long-distance migration. Therefore, we can refer to locust migration to study the migration of *O. decorus asiaticus*. Moreover, previous studies found that *O. decorus asiaticus* has migration behavior, and temperature and wind can significantly affect the flight of *O. decorus asiaticus* at night [19]. *O. decorus asiaticus* can fly 150 km per night with the help of wind [19]. Beijing is about 600 km away from Mongolia, so *O. decorus asiaticus* in Mongolian can reach Beijing in less than five nights. Through light trapping and trajectory simulation analysis, it is found that *O. decorus asiaticus* mostly migrate downwind at night, which is consistent with the flight behavior of the Senegalese *Oedaleus senegalensis* [43].

On 12 July 2002, it was reported that a large number of *O. decorus asiaticus* appeared in Beijing. Jiang Xiang et al. conducted a telephone survey and found that a large number of *O. asiaticus* appeared in Chengde City in Hebei Province, Zhangjiakou City in Hebei Province, and Beijing between 10 and 12 July 2002 [4]. Because most of the *O. asiaticus* distributed in China in early July were elderly nymphs [44], the source of the *O. asiaticus* is likely to be from abroad. Therefore, the wind direction at night at that time shows that grasshoppers followed the airflow from Mongolia into China and landed in Hebei province (Figure 3). In addition, according to the records of the Beijing Academy of Agricultural Sciences, there was a large-scale migration of *O. asiaticus* in Xisu Banner in Inner Mongolia, China, on 18 July 2003. The swarm entered Hohhot in China from 30 to 31 July, and a large number then entered Yanqing on 3 August. This was probably because the migratory *O. asiaticus* entering Xisu Banner were the Mongolian *O. asiaticus* migrating downwind and entering Chinese territory, leading to a large-scale migration of adult insects appearing (Figure 4). According to the analysis of the migration dynamics of *O. decorus asiaticus* in 2002 and 2003 (Figure 3 and Figure 4), we can summarize the possible migration routes of *O. decorus asiaticus* after entering China from Mongolia as follows: (1) entering Erlianhaote, Xianghuang Banner, Zhangjiakou City, and Hebei Province through the west Sunite Banner of Inner Mongolia, and continuing to Yanqing City, Beijing; (2) entering China via the Abaga Banner Xilingol league, Xilinhot in Inner Mongolia and continuing to move south to Yanqing. At the same time, we mainly analyzed the influence of the Mongolian cyclone center on the migration of *O. decorus asiaticus*. There are also extratropical cyclone centers in adjacent areas, such as Eastern Inner Mongolia and Heilongjiang [45,46]. However, combined with field observation data, there were no adult *O. decorus asiaticus* in these areas in the beginning and middle of July, which needs to be further studied.

## 5. Conclusions

*Oedaleus decorus asiaticus* is known from Russia (southern Siberia including the Tyva Re-public, the Republic of Buryatia, and Transbaikal region), the Mongolian People’s Republic, and the People’s Republic of China, but sometimes is considered as a subspecies of widely distributed *O. decoratus* [40,47]. *O. decorus asiaticus* is an important pest affecting the farming pastoral ecotone of northern China, which has caused great losses to agriculture and animal husbandry [3,4,44]. The results of the study showed that airflow is very important for the long-distance migration of *O. decorus asiaticus*, and that wind shear, in particular, was the main factor related to its forced landing. Analyzing the weather records, we found that the northwest wind prevailed when the population invaded. Specifically, from July to August, a large number of emerging adults appeared in the source areas of Mongolia, and the large-scale northwest wind was the decisive condition for the successful long-distance migration of *O. decorus asiaticus*. It has a strong migratory ability, flying along the airflow for several nights. If the northwest air current meets the southwest warm current going north, a large number of *O. decorus asiaticus* will drop due to wind shear, and then a major outbreak will occur. The analysis of the source of the insects shows that the *O. decorus asiaticus* break outs in China may have originated from Mongolia. They were brought into China by the southerly airflow at night, and they could make a forced landing in Beijing due to wind shear, sinking airflow, rainfall and other reasons. In sum, through the analysis of the insect prevalence and meteorological conditions in Mongolia, we can provide a basis for predicting the occurrence of *O. decorus asiaticus* in China, and the possible migration routes of *O. decorus asiaticus* after entering China from Mongolia are as follows: (1) entering Erlianhaote, Xianghuang Banner, Zhangjiakou City, and Hebei Province through the west Sunite Banner of Inner Mongolia, and continuing to Yanqing City, Beijing; (2) entering China via the Abaga Banner Xilingol league, Xilinhot in Inner Mongolia and continuing to move south to Yanqing. Both *O. decorus asiaticus* and *O. senegalese* can migrate downwind to other areas at night [6,48]. Therefore, In the main invasion season in summer, we should make full use of high-altitude searchlights, insect radar and other pest monitoring technologies to improve monitoring and early warning of *O. decorus asiaticus* appearances in northern China.

## Figures and Tables

**Figure 1 insects-13-00072-f001:**
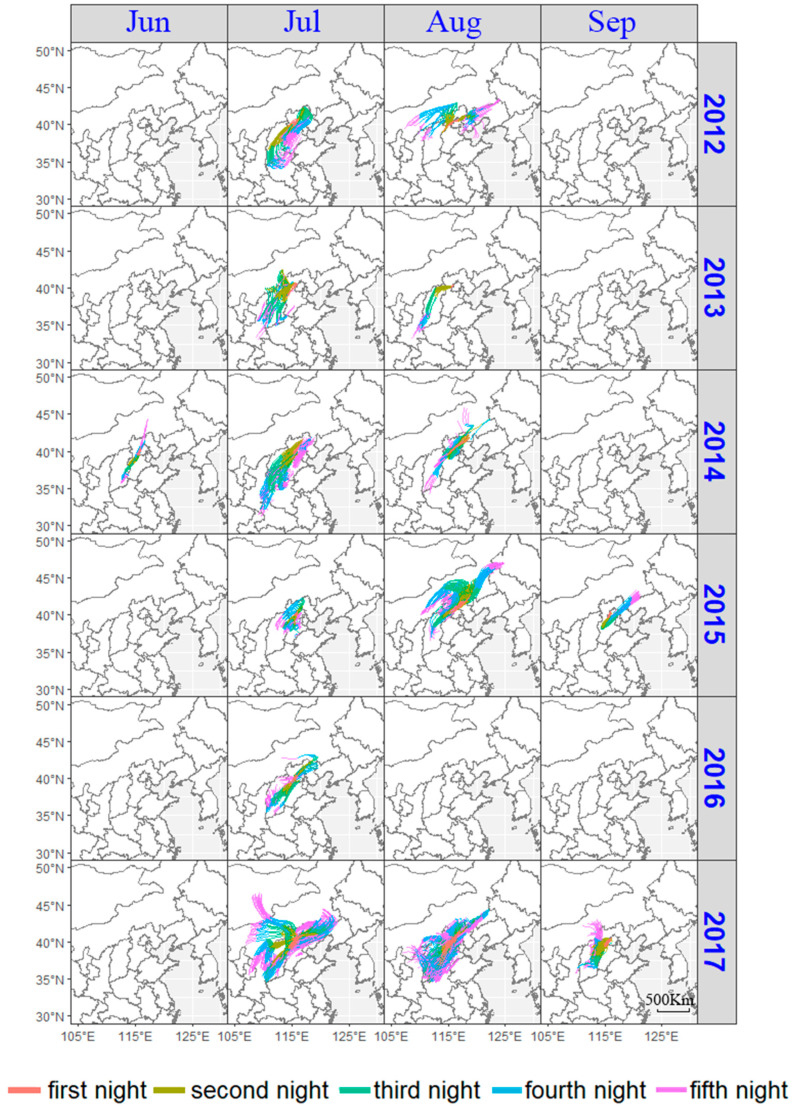
Backward trajectories from Yanqing in Beijing between 2012–2017.

**Figure 2 insects-13-00072-f002:**
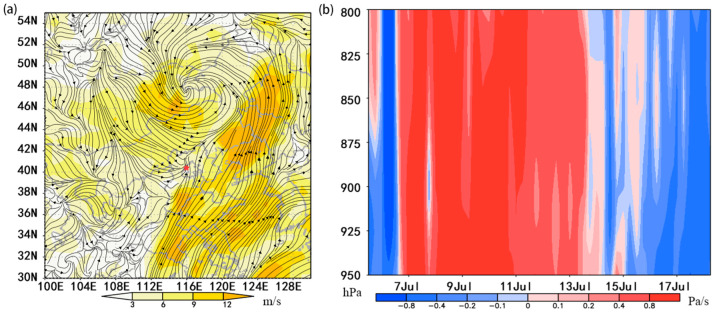
(**a**) Horizontal wind field (m s^−1^) on 925 hPa on the night of 8 July 2017 in Yanqing; (**b**) vertical velocity (Pa s^−1^) on 800–950 hPa on 7 July to 17 July 2017 in Yanqing.

**Figure 3 insects-13-00072-f003:**
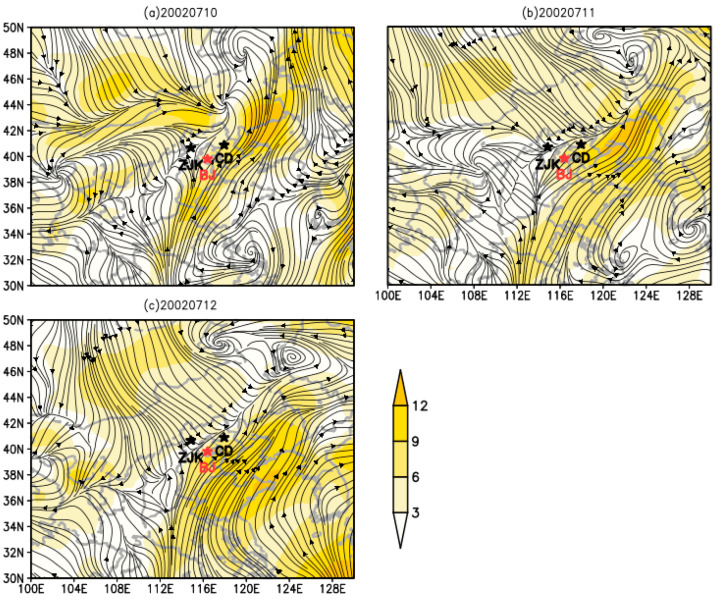
Night wind fields at 925 hPa during the outbreak of the *Oedaleus decorus asiaticus* in Beijing. Color scale bar indicates wind speed in m s^−1^ (note: 1. CD: Chengde, ZJK: Zhangjiakou, BJ: Beijing; 2. (**a**) 20020710: 10 July 2002; (**b**) 20020711: 11 July 2002; (**c**) 20020712: 12 July 2002).

**Figure 4 insects-13-00072-f004:**
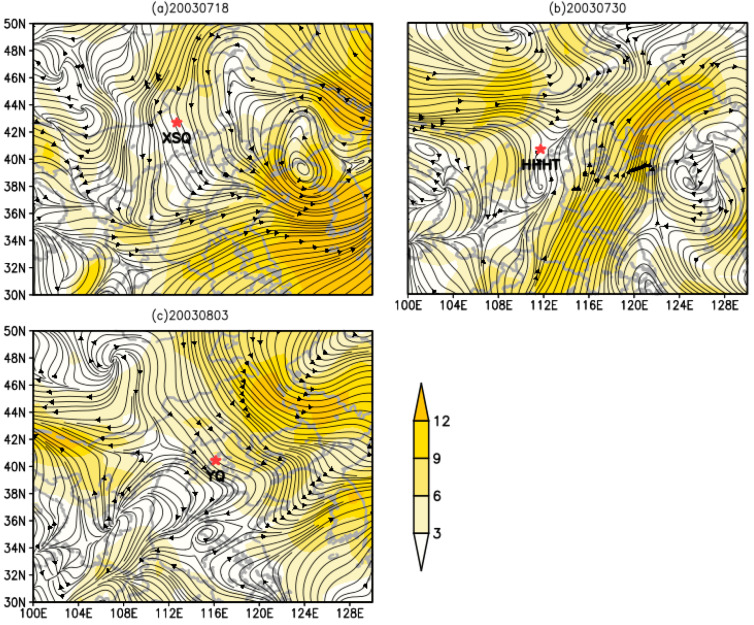
Wind fields at 850 hPa, Color scale bar indicates wind speed in m s^−1^ (note: (**a**) night wind field in Xisu Banner; (**b**) night wind field in Hohhot; and (**c**) night wind field in Yanqing).

**Table 1 insects-13-00072-t001:** Selection of the scheme and parameters of WRF.

Item	Domain1
Location	40° N, 116° E
The number grid points	99 × 99
Distance between grid points (km)	30
Layers	30
Map projection	Lambert
Microphysics scheme	WSM6
Longwave radiation scheme	RRTMG
Shortwave radiation scheme	RRTMG
Surface layer scheme	Monin-Obukhow
Land/water surface scheme	Noah
Planetary boundary layer scheme	YSU
Cumulus parameterization	Tiedtke
Forecast time	72 h

**Table 2 insects-13-00072-t002:** The daily density of *Oedaleus decorus asiaticus* collected by trap in Yanqing from 2012–2017.

Year	Date of First Capture		Date of Final Capture		Duration (d)	Date of Peak Catches (n)		Time of Maximum Capture	Total Catches
	Number (n)	Time	Number (n)	Time		Number (n)	Time		
2012	19 July (1)	20:00–21:00	23 August (1)	01:00–02:00	36	20 July (4)	20:00–21:00	20:00–21:00	11
2013	8 July (1)	22:00–23:00	24 August–25 August (1)	24:00–01:00	49	16 August (4)	20:00–21:00	20:00–21:00	21
2014	30 June (1)	22:00–23:00	19 August (1)	21:00–22:00	52	19 August–20 August (7)	24:00–01:00	24:00–01:00	32
			19 August–20 August (9)	23:00–01:00					
2015	13 July (1)	21:00–22:00	4 September (2)	20:00–22:00	53	15 August (3)	20:00–21:00	20:00–21:00	34
						15 August–20 August (8)	22:00–02:00		
2016	27 July (1)	01:00–02:00	30 July (1)	02:00–03:00	4	-	-	-	2
2017	8 July (3)	21:00–23:00	11 September (2)	20:00–21:00	66	15 August–16 August (141)	20:00–02:00	20:00–21:00	566
						16 August (1)	04:00–05:00		

**Table 3 insects-13-00072-t003:** The backward trajectories of the *Oedaleus decorus asiaticus* in Yanqing.

Duration	Total	Valid Trajectories	Overrange Trajectories
1	672	539	133
2	4312	3210	1102
3	3210	3083	127
4	3083	3020	63
5	3020	2967	53
Total	14,297	12,819	1478

**Table 4 insects-13-00072-t004:** The endpoints distribution of the backward trajectories of the *Oedaleus decorus asiaticus* from 2012 to 2017.

Year	2012		2013		2014		2015		2016		2017	
Trajectories	Valid Trajectories	Overrange Trajectories	Valid Trajectories	Overrange Trajectories	Valid Trajectories	Overrange Trajectories	Valid Trajectories	Overrange Trajectories	Valid Trajectories	Overrange Trajectories	Valid Trajectories	Overrange Trajectories
Inner Mongolia	83	1	24	3	43	7	546	23	15	3	1091	12
Beijing	78	12	1	6	169	33	68	27	17	7	981	178
Ningxia	0	0	0	0	0	0	0	0	0	0	0	0
Jilin	5	0	0	0	1	0	4	0	0	0	0	0
Tianjin	7	0	0	0	9	0	11	0	0	0	12	0
Shandong	3	0	0	0	0	0	0	0	0	0	6	0
Shanxi	139	21	193	9	336	38	142	3	146	11	2156	113
Hebei	444	78	42	52	598	90	583	157	79	20	3722	554
Henan	39	0	9	0	93	0	0	0	0	0	282	6
Hubei	0	0	0	0	10	0	0	0	0	0	0	0
Chongqing	0	0	0	0	4	0	0	0	0	0	0	0
Shaanxi	5	0	70	0	33	0	0	0	12	0	137	0
Liaoning	83	0	0	0	0	0	34	0	4	0	150	0
Heilongjiang	0	0	0	0	0	0	39	0	0	0	0	0
Gansu	0	0	0	0	0	0	0	0	0	0	20	0
Mongolia	0	0	0	0	0	0	0	0	0	0	49	0
Sea	4	0	0	0	0	0	0	0	0	0	2	0
Total	890	112	339	70	1296	168	1427	210	273	41	8608	863

## Data Availability

The data presented in this study are available in Appendix A.

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
