# Peer review of "An Analysis of the Possible Migration Routes of Oedaleus decorus asiaticus Bey-Bienko (Orthoptera: Acrididae) from Mongolia to China"

_insects, 2022, doi:10.3390/insects13010072_

Round 1
Reviewer 1 Report
Dear authors and editors,
I believe this manuscript is very interesting and important not only for acridologists, but for the wide audience of entomologists (including the applied ones) and ecologists as well. However, the actual versions of the text includes some serious flaws. Besides, there are some technical problems and typos. This is why I recommend the major revision.
I guess that there are several main problems:
(1) To separate two subspecies of Oedaleus decorus is not good idea. In 1981 Ritchie revised this genus and showed that O. asiaticus is conspecific with O. decorus. Up to date there are no evidences that O. decorus asiaticus may be the separate species (cf., e.g., some comments [Sergeev et al., 2020]).
(2) In the beginning and in the end of the text, the authors hypothesize that the main sources of migrating groups of O. d. a. are in Mongolia. However, their results don't support strongly this idea, and, in the main text, the authors discuss the possible routes of the species migrations in more or less accurate manner.
Actually there are several important issues:
(2.1) There are more than 500 km between Mongolia and Beijing and its vicinities.
(2.2) There are some short, but relatively high mountain ranges often with altitudes more than 1,500 m a.s.l.
Both points (2.1 and 2.1) show that some possible routes from Mongolia to the Beijing area are limited by distance and barriers.
(2.3) The vast area between Beijing and Mongolia is mainly occupied by the grasslands and the semi-deserts where numerous and abundant populations of O.d.a. may exist. In this case, according Occam's razor the simplest explanation will be to associate migrating adults of O.d.a. with the grasslands of Inner Mongolia.
(2.4) The authors declare that the stable populations of the species may not exist near Beijing. I believe that they can. Perhaps, the natural conditions near Beijing are not very optimal for this species, but they allow the species to exist in local landscapes, at least, in applicable habitats.
(2.5) The authors try to use some observations concerning grasshopper development and described the following pattern: at least in some cases, when the adults of O.d.a. landed and suddenly became abundant, the local colonies of the species included only hoppers. But one should understand that, generally speaking, the climate of Mongolia is relatively cold. For instance, the mean monthly temperatures in Hohhot are 16.6 (May) and 21.3 (June), in Xilinhot — 13.5 and 19.2. In Sainshand (Mongolia) — 14.5 and 20.4 accordingly. This means, in Mongolia, O.d.a. develops either in the same time as in Inner Mongolia or later.
(2.6) O. decorus asiaticus is not the so-called true locust. The species is the grasshopper with some trends toward gregarization [Cease et al., 2010]. The similar patterns were described for Oedaleus senegelensis [Cheke, 1990; Farrow, 1990; Maiga et al., 2008; Elamin et al., 2013]. Both members of the genus are commonly characterized by nocturnal flights over relatively short daily distances with landing in the end of nights or in mornings and take-offs in evenings. This means that such grasshoppers may migrate over relatively long distances (about several hundred kilometers) but "from stone to stone", and applicable conditions for take-off and landing should be on each stage of their dispersals.
(2.7) The authors discuss only the Mongolian cyclone center, but in adjacent territories, there are several other extratropical cyclone centers (e.g., in the eastern part of Inner Mongolia, in Heilongjiang etc. [Yang et al., 2018; Antokhina et al., 2019; Lee at al., 2020]). They may be also important for the species.
(3) I guess this will be better if the authors will use data for Oedaleus senegalensis (not for the locusts) and compare their results with the results obtained for this species.
Editorial and technical proposals and comments:
(4) Please, try to change the title, e.g.,
An analysis of the possible migration routes of the grasshopper Oedaleus decorus asiaticus (Bey-Bienko) (Orthoptera: Acrididae) in the eastern part of its range
line 24 — south > southward
lines 36–37 — in recent years — why? The problems with the Migratory locust and some grasshoppers were and are in China during millennia (e.g. [Chen, 1999; Zhang, Hunter, 2017; Bello, 2018]).
lines 37–40 — "Oedaleus asiaticus (O. asiaticus) is an important pest affecting grassland and adjacent farmland in northern China, and has damaged the ecological environment of the grasslands and caused huge damages to the livestock industry and herdsmen's livelihood." — please, try to simplify this sentence.
line 42 and so on — (1) > [1]
— please, also differ hyphens and en-dashes in all cases!
line 45–46 — "we found that the local population only at the 5th instar nymph stage (2), the latter can not migrate in" > we found that the local population consists only of the fifth stage hoppers [2[, the latter can't migrate over...
lines 50–62 — In 1990 Roger Farrow published the special review concerning locust dispersal and flights!
lines 75–76 and so on — please, remove the full stops in the end of Subsections
lines 79–80 — why there are different numbers of digits in the geographic coordinates? Really the forth digit is about 11 meters (along a meridian) and about 8–9 m (along this parallel)... May be two digits are enough?
lines 100–106 — There are two almost equivalent sentences... — please, remove one...
lines 141–143 — "Only when the ground height is lower than the 8 sea level altitudes used did the flight trajectory continue for five consecutive nights." — please, rewrite.
line 151 — "the grasshoppers take off at 8pm and landed at that time did not immigration" — please, rewrite.
Table 2 — clover cutworm?
Line 169 — "Analysis of the origin of O. asiaticus" > An analysis of the possible origin of migrating adults (or similar) (you don't discuss the origin of the species)
Line 170–171 — the high level of egg mortality is the typical and common situation for grasshoppers. Each female usually produces from several dozen to 120–150 eggs (depend on species). In the temperate and subtropic regions, the majority of embryos dies during winter and spring.
Line 188 — Augest > August
Lines 192–193 — "entering Hebei through Mongolia" — please, explain
Line 194 — How validity is determined?
Table 5 — I suggest to move this table to the Supplementary parts of the publication. It is very long and non-informative. Generally speaking, the authors may try to summarize data on a Figure.
lines 239–241 — "On 8 July 2017, pressure in the Yanqing 239 District was at a height of 925hPa was affected by the Mongolian cyclone" — please, rewrite.
Part 4 — I suggest to compare your results with results obtained for O. senegalensis. Such comparisons allow to describe the situation in more explicit manner.
line 303 — damage — what types of damages? Agricultural fields, pasturelands, orchards...
line 304 — "long-distance migration" — actually there are no real evidence for very long dispersal of O.d.a. The authors (and authors of several other publications) hypothesize only that they are.
line 307 — the source > the possible source
lines 309–310 and so on — please, use italics for the formal names of species.
line 368 — sheer > shear (?)
line 370 — "Dynamic analysis of O. asiaticus migration in 2002 and 2003, we can summarize..." — please, rewrite
lines 390–396 — some other routes (e.g., from Inner Mongolia) may be used as well. I believe that the authors can designated several different opportunities for the species migrations and some of them are shorter and look like more reasonable.
Figures: please, add scales for all maps
and unify explanations (e.g., m/s — Fig. 2 and m s-1 — Fig. 3 (and should be with superscripts in the last case))
References: please, rearrange them according the Journal rules and add doi (where that is possible).
Author Response
We thank the anonymous reviewer for his/her encouragement and marking ‘yes’ for most of the qualitative attributes mentioned above. For the result and conclusions section has been improved to the best of our ability and understanding.
Please see the attachment.

Reviewer 2 Report
Comments of reviewer
- Headline of the Table 2 “The daily density of clover cutworm collected by trap in Yanqing from 2012-2017” is incorrect, because clover cutworm is a moth Scotogramma trifoli from order Lepidoptera. Probably O. asiaticus is correct?
- There are a lot of typing errors in text, for example:
Line 259 – “O. asiaticus” must be Italic;
Line 336 – “in China.The” the gap is missing;
Line 344 – “example, Guan Jiang cheng et al. analyzed” must be “example, Guan Jiang Cheng et al. analyzed”.
Therefore the corrections of text by native English speaker are in need.
- Lines 347-349, printed:
Bao Yunxuan et al. analyzed the migration process of Sogatella furcifera that appeared on July 10-11, 2003 and found that rainfall was the direct cause of the concentrated landing of S.furcifera (34). (Latin name must be Italic).
replace for:
Bao Yunxuan et al. analyzed the migration process of Sogatella furcifera that appeared on July 10-11, 2003 and found that rainfall was the direct cause of the concentrated landing of this planthopper species (34).
- The correct Latin name and taxonomic status of Oedaleus asiaticus are nor clear, it is considered as separate species or as subspecies of O. decoratus (see Sergeev et al, 2020, page 19 for details); the PDF of this paper may be easy found by DOI: 10.25221/fee.402.1
To discuss the possible migration of O. asiaticus (if you believed that it as a separate species) from the Mongolian People’s Republic directly to Beijing in China the data on total distribution of this species must be given at least in “Conclusion” as follow (Lines 377-378):
Oedaleus asiaticus is known from Russia (southern Siberia including the Tyva Republic, the Republic of Buryatia, and Transbaikal region), the Mongolian People’s Republic, and the People’s Republic of China, but sometimes is considered as a subspecies of widely distributed O. decoratus (37-38). O. asiaticus is an important pest affecting the farming pastoral ecotone of northern China, which has caused great losses to agriculture and animal husbandry (1-2, 36).
In this case add to Reference list after Line 507 follow papers:
- Ritchie, J.M. 1981. A taxonomic revision of the genus Oedaleus Fieber (Orthoptera: Acrididae). Bulletin of the British Museum (Natural History). Entomology, 42(3): 83–183.
- Sergeev M.G., Storozhenko S.Yu., Benediktov A.A. 2020. An annotated check-list of Orthoptera of Tuva and adjacent regions. Part 3. Suborder Caelifera (Acrididae: Gomphocerinae: Gomphocerini; Locustinae). Far Eastern Entomologist, 402: 1-36.

Author Response
We really appreciate the keen interest, fruitful suggestions, appreciation and encouragement by the honorable reviewer. To the best of our knowledge, the revised version in the light of the comments of the reviewer has been greatly improved.
Please see the attachment.

Reviewer 3 Report
This manuscript presents interesting results and explanation how Oedaleus asiaticus, an important grasshopper pest in Asia, migrates. The authors used light trap data and simulations based on weather forecast model. The manuscript fits very well into Insects scope, is generally well written and the results presented can have implications for improving pest management strategies. There are several points which should be improved/clarified before the manuscript is accepted for publication. For more details see comments below (numbers indicate lines in ms).
Specific comments
Title: Please add either „grasshopper” and/or „(Orthoptera: Acrididae)“ so that reader not familiar with this species knows where the species belong to.
Besides abstract new submissions to Insects should also have “Simple summary” before the abstract (please check current Insects template or instructions for authors.
Abstract
18 – instead of “etc.” it would be better to be more specific, e.g. “and other countries in Asia” or something similar.
Keywords: since species name is in the title it is usually not necessary to repeat here so “grasshoppers” could be better; use lower case in meteorology, optionally other keywords like “long distance flight, air current, weather model” could be added
Introduction
Background is sufficiently overviewed and aim of the study is logic and clear. Perhaps some literature on pest status/damage inflicted and how this pest can be controled could be inserted in the the beginning of first paragraph.
42 onwards: Please check citation format, I think is should be [1].
42 had > were
45-46 is seems that verb is missing in this sentence
51 onwards: species names should be in italics
Materials and Methods
76 subsubsection title should not be italics
81-90 this part needs improvement as not all is clear: first it is written that the trap lamp was turned on a 8 p.m and is replaced every 1 hour while later is written “searchlight are turned on at sunset ...” and that nylon net bag was replaced very two hours. I also suggest to use past tens and describe light trap construction first and than when and how it was operated. Was any chemical like chloroform used to anesthetize insects as it is usual in common light traps?
104-406 this is redundant as is already written in the beginin of paragraph.
119 I suggest to refer to Figure 1 in Results, not here.
143-146 Specification of the WRF model should be moved to section 2.2.1.
151 “did not immigration” > “did not immigrate”
Results
155 – terms should be consistent in the ms so I suggest to use light traps instead, the sentence needs linguistic correction.
161 – this sentence is not clear, should it mean “There were almost no light-captured grasshoppers in other months”?
166 Table 2 caption – clover cutworm? Certainly a mistake. Please specify that it was collected by light trap.
170 It is recommended that full genera is written when it is in the beginning of sentence, i.e. not abbreviated.
170-171 this fits better into Discussion, indeed, no references should be used in Results.
200 Figure 1 – I suggest to enlarge this figure a bit as it is rather difficult to see all details. Moreover, some colors look the same.
209 please leave references for Discussion
255 – y-axis labels are missing in graphs
263 – more appropriate for Discussion
266 – worms should probably read “swarms”
281 - species name should be in italics
282 – upper index in s-1 should be used
299 – the same as in 282
Discussion
302 – Do not start sentence with abbreviated species name.
303-304 It would be nice to support this by few references.
309 and onwards – use italics for species names
348 – it is always good to add name of author who described species and also insect order and family when the species is mentioned for the first time in ms.
362 - with the rainy days?
370 - needs to be reformulated, should it mean like “Based on dynamic analysis ….”
Conclusion
This is too long to me, I suggest to shorten it and remove the first sentence which is not based on results of the present study (also there should not references in conclusions). Some information on two migration routes is repetition from end of Discussion and should be mentioned only once (better in Conclusion).
Author Response

(The authors gave the same response as above.)

Round 2
Reviewer 1 Report
Dear authors,
Thanks for your great efforts to improve the text. However, I suggest to check it once again.
(1) line 80–81 — I am almost sure that the species is in this area (but in natural environment, it may be very rare). In this context, I propose to use more or less diplomatic construction.
(2) Please, differ hyphens and en-dashes in all cases!
line 73 and so on [11-14] > [11–14]
and in the References as well
1175-1182 (and so on) > 1175–1182
(3) some typos:
line 47 — had 2 > had two
line 57 and so on — [1-2] > [1,2]
line 196 and so on — please, remove the full stops in the end of Subsections
Table 2 — asiaticu > asiaticus
line 313 — (stal) > (Stål)
line 416 — Re-public > Republic
line 561 — Oedaleus — in italics
line 563 — Saussure > Saussure
Figures: please, unify explanations (e.g., m/s — Fig. 2 and m s-1 — Fig. 3, 4 (and should be with superscripts in the last case))
